# MindX: A Low-Cost Benchmark System for Real-Time Brain-Controlled Gaming

## Abstract

We introduce **MindX**, an open benchmark platform for real-time EEG-based game control using low-cost, consumer-grade hardware. The system features three directional-control games—*Hide-and-Seek*, *Rhythm Game*, and *Snake Game*—designed to elicit motor imagery and attention-related neural signals. A lightweight CNN+LDA model processes 4-channel EEG from a Muse S headset and issues directional predictions (left and right) every 300 ms, with end-to-end latency under 350 ms. A user-centered co-adaptive loop enables lightweight personalization based on gameplay feedback. In a pilot study with three users, MindX achieved 76% accuracy, well above the 50% baseline. The framework provides a reproducible and extensible testbed for evaluating real-time EEG decoding pipelines.

## 1 Introduction

Brain-computer interfaces (BCIs) enable control of digital systems from neural activity, supporting applications in assistive technology Kosmyna & Maes (2019), neurorehabilitation, and entertainment Heim et al. (2025). Electroencephalography (EEG) is the most common sensing modality due to its non-invasive, portable nature, yet state-of-the-art EEG systems still rely on clinical-grade hardware and offline datasets, limiting real-time deployment, scalability, and reproducibility Jayaram et al. (2018).

Recent deep learning advances, particularly transformer-based models Zhao et al. (2024); Zhang et al. (2023), have improved EEG decoding by capturing long-range temporal patterns, but few are adapted for lightweight, real-time use with consumer devices such as Muse or OpenBCI. Existing BCI toolkits often target binary tasks (e.g., SSVEP spellers) and lack modular pipelines for reproducible, interactive decoding.

We present **MindX**, the first open benchmark system for real-time, multi-directional EEG game control on low-cost, mobile hardware. MindX features three cognitively diverse games—*Hide-and-Seek* (reaction), *Rhythm* (attention), and *Snake* (continuous control)—a lightweight CNN+LDA hybrid classifier, and a duo-stage personalization loop for unsupervised feedback-driven adaptation. Running on the Muse S headset (4 frontal electrodes Sawangjai et al. (2019)), the pipeline produces left/right predictions every 300 ms with end-to-end latency of ∼300–400 ms. In a pilot study with three participants, accuracy reached 72–76%, well above the 50% chance baseline. Developed with a user-centered co-design approach, MindX integrates post-game feedback, unsupervised calibration, and open-source release for community-driven BCI research.

**Our contributions are as follows:**

- We introduce **MindX**, an open benchmark suite of three EEG-controlled games supporting real-time, multi-directional control.

- We develop a modular signal-to-action pipeline that integrates denoising, CNN+LDA modeling, and a lightweight co-adaptive personalization loop.

- We evaluate the system with real users, demonstrating low-latency online decoding and directional accuracy significantly above baseline.

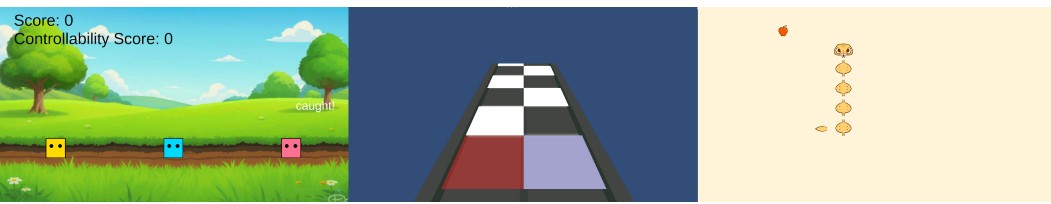

Figure 1: Screenshots from the MindX benchmark suite: **Left:** Hide-and-Seek (reaction), **Middle:** Rhythm (attention), **Right:** Snake (continuous control). The games evoke distinct cognitive signals, providing a diverse testbed for EEG decoding.

GAMES AND HARDWARE

The MindX suite includes three interactive games—*Hide-and-Seek* (reaction), *Rhythm* (attention), and *Snake* (continuous control)—that elicit distinct cognitive features such as frontal $\mu$ suppression, ERP-like transitions, and alpha modulation. Together they provide a diverse testbed for EEG decoding (Figure 1). The system runs on the Muse S headset, a consumer-grade device with 4 frontal electrodes and Bluetooth streaming Sawangjai et al. (2019), enabling mobile, non-clinical deployment.

Table 1: Mapping of game tasks to cognitive domains and EEG signal characteristics.

| Game | Cognitive Task | EEG Domain | Expected Feature |
|------|----------------|------------|------------------|
| Hide-and-Seek | Motor Imagery (Left/Right) | Frontal $\mu$ suppression | Channel lateralization |
| Rhythm Game | Directional Intent | Temporal planning focus | ERP-like transition pattern |
| Snake Game | Sustained Attention | Frontal alpha modulation | Hemispheric asymmetry |

## 2 RELATED WORK

EEG-BASED BRAIN-COMPUTER INTERFACES AND DEEP LEARNING

Traditional EEG decoding relied on handcrafted features and classical classifiers, but deep learning models now dominate for their robustness and generalization Schirrmeister et al. (2017). Benchmarking platforms such as MOABB Jayaram et al. (2018) and PhysioNet Goldberger et al. (2000) provide standardized datasets, yet mainly focus on offline, clinical, or high-density recordings. Transformer-based models capture long-range temporal dependencies and have shown strong performance in EEG decoding, as in CTNet Zhao et al. (2024), LocalFormer Zhang et al. (2023), and EMPT Liu et al. (2024). Cross-modal extensions (e.g., MindPainter Yu et al. (2025), NeuralFlix Sun et al. (2025)) highlight the potential of attention mechanisms, though evaluations remain restricted to lab-grade, high-channel EEG, limiting real-time applicability.

LOW-COST EEG SYSTEMS AND REAL-TIME DECODING

Recent work has emphasized low-cost, consumer-grade EEG systems for more accessible BCI development. Muse InteraXon Inc. (2025) and OpenBCI devices enable mobile EEG acquisition, but reproducible pipelines for real-time control remain rare. Prior work by Heim et al. (2025) and Kosmyna et al. Kosmyna & Maes (2019) explored co-adaptive feedback loops using consumer headsets, showing that simple personalization can boost decoding stability. Zhao et al. Zhao et al. (2025) and Su et al. Su et al. (2022) proposed efficient hybrid CNN-transformer models to decode motor intention with high frequency, but lacked integration into open-source game control platforms.

While some studies evaluated online feedback Xu et al. (2022), very few offer open datasets, benchmark tasks, or toolkits for end-to-end real-time BCI gameplay. Most focus on binary paradigms (e.g., SSVEP/P300 spellers), limiting the study of directional or continuous control.

ADAPTIVE PIPELINES AND FEEDBACK-DRIVEN LEARNING

To address variability across users and sessions, adaptive strategies have been explored. Co-learning and real-time feedback Kosmyna & Maes (2019); Heim et al. (2025) reduce calibration demands, while clustering and pseudo-labeling support continuous adaptation Chang et al. (2024); Chen et al. (2024). Lightweight mechanisms such as PCA+KMeans calibration Chang et al. (2024) show promise for real-world scenarios, enabling improvement without ground-truth labels

GAP AND CONTRIBUTION

Despite these advances, there remains no open, reproducible, community-driven benchmark for real-time, multi-directional EEG game control using consumer-grade devices. Existing systems are typically limited to binary tasks, non-public code, or offline evaluation.

To address this gap, we present **MindX**, a modular benchmark platform supporting real-time directional control across cognitively diverse games using low-cost EEG hardware. Our system integrates a CNN+LDA hybrid decoder with a duo-stage personalization loop and supports plug-in extensibility. All components will be made publicly available to support reproducible, real-world EEG research.

## 3    SYSTEM DESIGN: MINDX

MindX is a modular benchmark system for real-time EEG-based game control using low-cost, consumer-grade EEG headsets. Its architecture comprises three main stages: (1) EEG acquisition via a mobile device, (2) lightweight signal processing and classification, and (3) real-time game interaction and feedback adaptation. The system is designed for extensibility, reproducibility, and online personalization across multiple game genres and user types.

To promote adoption and experimentation, each component is built as an independent module with defined interfaces, enabling researchers to substitute models, adapt signal processing methods, or plug in new games with minimal modification.

### 3.1    EEG ACQUISITION

The system uses the Muse S headset, a commercially available, Bluetooth-enabled EEG device with four frontal electrodes (TP9, AF7, AF8, TP10) and a sampling rate of 256 Hz. This low-cost, portable setup enables untethered use in non-clinical environments while maintaining sufficient signal quality for motor imagery decoding. The frontal electrode configuration follows prior BCI protocols Lotte et al. (2018); Blankertz et al. (2007), making it compatible with standard decoding techniques.

To support real-time performance, EEG streaming is implemented using the Lab Streaming Layer (LSL), ensuring low-latency data capture. The system runs on standard laptops without dedicated GPUs, demonstrating its accessibility for classroom or at-home experimentation.

### 3.2    EEG-GAME BENCHMARK SUITE

MindX includes three cognitively distinct EEG-controlled games, designed to elicit varied neural activity patterns and challenge different decoding capabilities:

- **Hide-and-Seek:** A directional game driven by left/right motor imagery.

- **Rhythm Game:** A rapid lane-changing task requiring imagined directional intent.

- **Snake Game:** A focus-and-react game that emphasizes sustained visual attention.

Each game is mapped to a specific EEG domain—motor imagery, planning, or attention—and is expected to activate characteristic signal patterns (Table 1). The diversity of game mechanics supports cross-task generalization and enables multi-domain BCI evaluation.

To reduce interface learning effort, all games use the same directional control vocabulary (left and right) and a similar visual feedback design. Task difficulty and timing constraints can be adjusted dynamically on Unity engine for future training or adaptation studies.

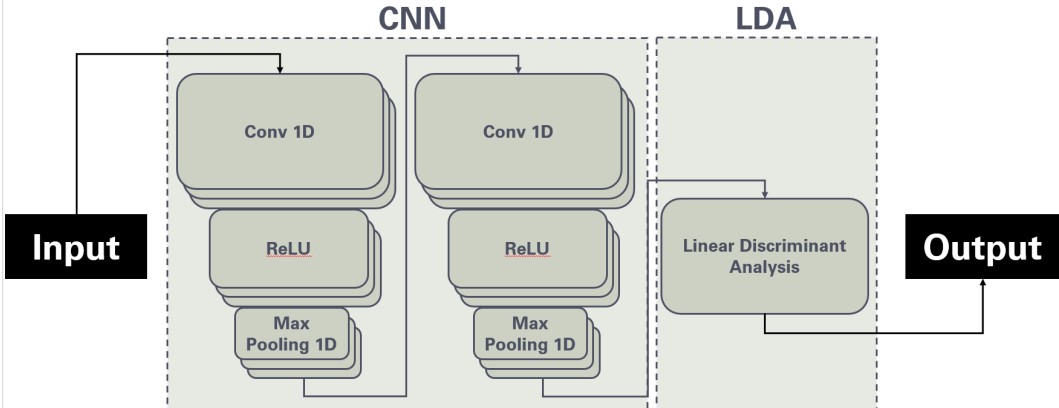

Figure 2: Overview of the proposed CNN–LDA architecture. The CNN extracts discriminative temporal features from EEG signals, which are then classified using LDA to maximize inter-class separability.

### 3.3 CNN+LDA MODEL ARCHITECTURE

We employ a hybrid architecture that integrates a convolutional neural network (CNN) as a feature extractor with a linear discriminant analysis (LDA) classifier, as shown in Figure 2. The CNN processes EEG inputs through two 1D convolutional blocks (with 8 and 16 filters, respectively), each followed by batch normalization, ReLU activation, and max pooling. This design captures local temporal dependencies while progressively reducing noise and dimensionality. The extracted representations are then flattened and projected into a 16-dimensional dense layer, with $L_2$ regularization applied to mitigate overfitting.

Instead of a softmax layer, we adopt LDA for classification. LDA maximizes inter-class separability while constraining intra-class variance, offering robustness and interpretability particularly suited for EEG signals. This hybrid approach combines the CNN's ability to learn discriminative temporal features with the statistical efficiency of LDA, yielding a compact yet effective model for EEG classification.

### 3.4 DUO-STAGE PERSONALIZATION PIPELINE

MindX also integrate a duo-stage personalization pipeline (see Section 4.8) and system architecture in Appendix Figure 4.

### 3.5 MODULARITY AND REPRODUCIBILITY

All modules are implemented with clean APIs and will be open-sourced for reproducibility. Our modular framework allows researchers to plug in and benchmark different models (e.g., CNN, Transformers, etc.), ensuring transparency and enabling future extensions.

## 4 METHODOLOGY

MindX is an end-to-end framework for decoding directional intent from raw EEG in real time with consumer headsets. Its modular design supports multiple classification models, with the hybrid CNN+LDA pipeline adopted as our final architecture. The following sections describe this design and the baselines used for comparison.

## 4.1 EEG Signal Preprocessing

Raw EEG signals are sampled at 256 Hz from four frontal channels (TP9, AF7, AF8, TP10). To reduce high-frequency noise while preserving temporal structure, we apply a causal moving average filter:

$$\tilde{x}_t = \frac{1}{w} \sum_{i=t-w+1}^{t} x_i \tag{1}$$

where $w = 5$ controls smoothing sharp fluctuations without introducing significant delay. This preprocessing step improves model stability and reduces false activations.

## 4.2 Proposed CNN+LDA Pipeline

Given an EEG window $X \in \mathbb{R}^{T \times C}$, a 1D CNN encoder $f_\theta$ extracts temporal–spatial features:

$$z = f_\theta(X) \in \mathbb{R}^d. \tag{2}$$

During training, $f_\theta$ is supervised with a temporary softmax head, but at deployment the features $z$ are classified using Linear Discriminant Analysis (LDA):

$$\hat{y} = \arg\max_i \; \text{LDA}(z)_i. \tag{3}$$

This two-stage design leverages CNN feature learning and LDA's robust linear separation.

## 4.3 Baselines for Comparison

For benchmarking, we also implemented several end-to-end neural models.

**CNN/Transformer with Softmax.** We trained both CNN and Transformer encoders followed by a softmax classifier:

$$\hat{y} = \arg\max_i \; \text{softmax}(Wz + b)_i. \tag{4}$$

Training uses cross-entropy loss, with entropy-based regularization to improve robustness (see Appendix B.3). Details of the Transformer encoder and attention formulation are provided in Appendix B.1.

**Channel-Wise Attention.** For interpretability, we explored a channel-wise attention mechanism that assigns importance scores to electrodes:

$$\alpha_c = \frac{1}{T} \sum_{t=1}^{T} \text{softmax}(w^\top x_{t,c}), \tag{5}$$

producing a vector $\boldsymbol{\alpha} \in \mathbb{R}^C$ that highlights informative channels. Additional derivations and visualization methods are provided in Appendix B.2.

## 4.4 Thresholding and Control Filtering

In practice, a prediction is accepted as a valid directional command only if:

$$\max_i \hat{y}_i > \tau \quad \text{and} \quad \mathcal{H} < \delta \tag{6}$$

where $\tau$ and $\delta$ are empirical thresholds (e.g., $\tau = 0.65$, $\delta = 1.0$). For CNN+LDA, $\hat{y}_i$ denotes LDA posteriors, and for baselines softmax outputs; this gating ensures stable real-time control.

## 4.5 Real-Time Inference and Scheduling

The system issues directional predictions every 300 ms using a sliding 2s window over incoming EEG. Each inference cycle includes preprocessing, embedding, attention encoding, and classification. Average end-to-end latency is ~300 ms (Table 2).

Table 2: Latency by processing stage (n=30).

| Stage | Mean (ms) | Std Dev |
|---|---|---|
| Preprocessing | 1.70 | 0.43 |
| Classification | 98.74 | 21.31 |

## 4.6 WINDOW SIZE JUSTIFICATION

We evaluated 0.5s, 1s, and 2s window sizes. The 2-second window achieved the best tradeoff between latency and decoding stability. Shorter windows failed to capture sufficient temporal context, while longer ones introduced perceptual lag.

## 4.7 ONLINE PERSONALIZATION VIA FEEDBACK LOOP

To reduce calibration effort and adapt across sessions, we implement a duo-stage training scheme. After each game session, recent EEG windows $X_{\text{session}}$ are clustered via PCA and KMeans:

$$\mathcal{D}_{\text{new}} \leftarrow \text{KMeans}(\text{PCA}(X_{\text{session}})) \tag{7}$$

Each cluster is pseudo-labeled by similarity to prior class centers and used for online fine-tuning, gradually improving alignment with the user's strategies (see Appendix B.4 for full formulation).

To quantify stability, we define:

$$S_{\text{ctrl}} = 1 - \frac{1}{T} \sum_{t=2}^{T} \mathbb{I}[y_t \neq y_{t-1}] \tag{8}$$

where $S_{\text{ctrl}} \in [0, 1]$ measures intra-session output smoothness.

## 4.8 MODEL CONFIGURATION AND RUNTIME SETUP

We trained the CNN+LDA model and others with standard hyperparameters (see Appendix B.5 for full details). All experiments were run in real time on a consumer laptop using PyTorch and Lab Streaming Layer (LSL).

## 5 EXPERIMENTS AND RESULTS

We evaluate MindX in terms of classification accuracy, latency, gameplay success, and subjective usability. Our pilot study focuses on the feasibility of low-cost, real-time BCI control using directional intent and serves as an initial benchmark for future systems.

## 5.1 EXPERIMENTAL SETUP

The system was tested using the Muse S EEG headset (256 Hz, 4 channels) across three real-time EEG-controlled games. Each participant controlled an in-game agent via mental commands (left and right). Trials were structured with a 1s cue period, a 1s thinking window, and a system response phase.

We conducted 20 trials per game (4 games in total) per subject (3 subjects in total) across 3 subjects (totaling 240 trials). Each subject completed the same game order and received identical instruction scripts. All games were rendered in Unity engine and executed on the same laptop with Bluetooth streaming from the Muse S.

## 5.2 EVALUATION METRICS

We report the following metrics:

- **Directional Classification Accuracy**: Percentage of trials where the system correctly predicted the intended direction.
- **Per-Class Accuracy**: Class-specific precision, computed as:

$$\text{Acc}_c = \frac{1}{N_c} \sum_{i=1}^{N_c} \mathbb{I}(\hat{y}_i = y_i) \tag{9}$$

- **Latency**: Mean and standard deviation of end-to-end system latency from cue onset to action output.
- **Gameplay Score**: Game-specific outcomes such as objects collected or obstacles avoided.
- **User Ratings**: Post-session survey on engagement, control, and fatigue using standardized Game Experience Questionnaire (GEQ) and System Usability Scale (SUS) instruments.

## 5.3 PRELIMINARY RESULTS

The model achieved directional prediction accuracy is around 72.24-76.59%, substantially above the 50% baseline. Average latency was $\mu = 297.4$ ms ($\sigma = 20.9$ ms), which supports real-time feedback.

Table 3 shows the accuracy of each model, revealing that our designed model, CNN + LDA, performs the best (76.59%), followed by Random Forest (63.8%), CNN (61.2%), KNN (60.7%), and SVM (60.4%). We observe that basic machine learning models perform about the same as, or even better than, the classic deep learning models CNN and EEGNet.

Table 3: Classification accuracy across all models.

| Model | Classification Accuracy | Model | Classification Accuracy |
|---|---|---|---|
| **CNN+LDA** | **76.59%** | DeepConvNet | 57.9% |
| Random Forest | 63.8 | CNN+LSTM | 57.6% |
| CNN | 61.2% | ShallowConvNet | 52% |
| KNN | 60.7% | CSP+LDA | 51.3% |
| SVM | 60.4% | Transformer | 33% |
| EEGNet | 58.6% | | |

These asymmetries may reflect differences in cortical activation or limitations in headset positioning. Similar lateral imbalances have been observed in motor imagery studies using frontal electrodes Lotte et al. (2018). A lack of data in the dataset could also contribute to such issues.

## 5.4 GAMEPLAY PERFORMANCE

Table 4 summarizes gameplay outcomes, reported in task-specific units with a common controllability score (percentage of correct directional commands). Hide-and-Seek achieved the highest controllability, Snake was most challenging, and Rhythm produced high task scores but only moderate controllability.

Table 4: Average scores of three participants playing the games

| Game | Score | Controllability Score |
|---|---|---|
| Snake Game | 5 | 0.341 |
| Hide and Seek | 93.67% | 0.591 |
| Rhythm Game | 4010/8000 | 0.404 |
| TUX Racer | 4 | N/A |

## 5.5 LATENCY CONSISTENCY

Latency remained stable across 180 trials, with a mean of 297.4 ms (Table 2), validating the system's suitability for real-time interaction.

## 5.6 USER FEEDBACK AND USABILITY

Table 5 (in Appendix) summarizes subjective ratings. Hide-and-Seek received the highest SUS (94.5/100), suggesting high perceived control. The Snake game had the highest reported challenge and fatigue. Positive affect was strongest in the Rhythm game, indicating user enjoyment despite control limitations.

Participants noted that mental effort was higher for continuous movement games, and that clear visual cues improved performance.

## 5.7 ABLATION STUDY

We conducted controlled experiments to isolate the effects of three design choices:

- **Window Size**: 2-second windows outperformed 0.5s and 1.0s in accuracy and stability.
- **Architecture**: The CNN+LDA hybrid yielded 15.39% higher accuracy than CNN alone, and 43.59% above Transformer-only models.
- **Smoothing**: Applying a moving average filter ($w = 5$) reduced false positives by 12.6%.

These findings support the robustness of our preprocessing and modeling pipeline.

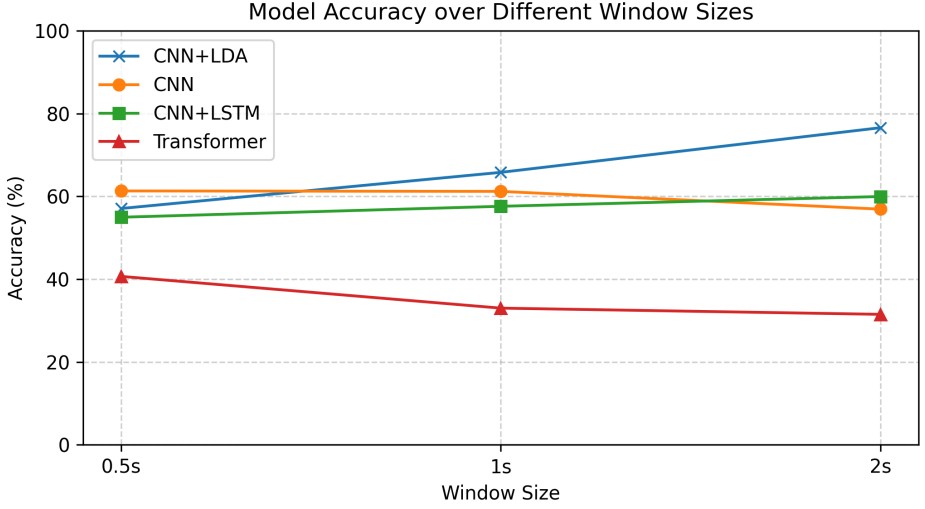

Figure 3: Model Accuracy over Different Window Sizes

## 6 DISCUSSION

Our preliminary findings demonstrate the feasibility of real-time, low-latency EEG-based directional control using a lightweight hybrid model and consumer-grade EEG hardware. The MindX framework supports diverse game genres and control paradigms, enabling broad applicability for interactive BCI research, education, and reproducible benchmarking.

## 6.1 EMPIRICAL FINDINGS AND USER EXPERIENCE

The 2-second EEG window offered the best tradeoff between temporal resolution and stability, aligning with prior transformer-based EEG studies Zhao et al. (2024); Liu et al. (2024). The CNN+LDA hybrid consistently outperformed standalone CNN or Transformer variants, highlighting the complementarity of spatial and temporal encoding Chang et al. (2024). Accuracy was higher for `right` commands, consistent with known lateral asymmetries in frontal EEG channels Lotte et al. (2018), and smoothing improved consistency in continuous tasks Su et al. (2022).

Participants reported stronger engagement and perceived control in discrete-response games (Hide-and-Seek, Rhythm), whereas continuous tasks (Snake) induced greater fatigue, consistent with prior findings on sustained-intention challenges in EEG interfaces Kosmyna & Maes (2019); Heim et al. (2025). Our duo-stage personalization loop yielded moderate gains without labeled retraining, resembling domain-adaptive fine-tuning Xu et al. (2022) but using unsupervised clustering for label propagation. Together, these results underscore both the technical feasibility and experiential constraints of real-time EEG control.

## 6.2 INTERPRETABILITY AND CHANNEL RELEVANCE

To understand which brain regions drive control decisions, we introduced a channel-wise attention scoring mechanism. This revealed consistent dominance of AF8 and TP10 (right hemisphere), particularly during `Right` intention trials. Such asymmetric activation is in line with known lateralization effects in motor imagery tasks Blankertz et al. (2007); Lawhern et al. (2018).

Incorporating interpretable attention patterns improves trust in BCI predictions and can inform future device designs with fewer, better-placed electrodes Roy et al. (2019). Integrating saliency maps Zhang et al. (2023) or gradient-based relevance methods could offer further insight into temporal-spectral contributions of EEG segments.

## 6.3 LIMITATIONS AND OPEN QUESTIONS

This pilot involved few participants with limited session diversity, raising concerns about generalization. Continuous tasks such as Snake showed control failures likely due to cognitive load, fatigue, and decoding delay, while robust artifact handling in non-clinical EEG remains an open challenge Sawangjai et al. (2019). We also note that all participants were highly familiar with the system, which may bias both usability feedback and decoding performance. Future studies should include new users and explore online artifact correction (e.g., ICA-free denoising Jiang et al. (2024)).

## 6.4 BROADER IMPACT AND APPLICATIONS

Beyond gaming, MindX's low-cost modular pipeline could support cognitive workload monitoring, neurofeedback, and educational tools Lotte et al. (2023), and may extend to assistive robotics for users with motor impairments when combined with zero-calibration transfer methods Wimpff et al. (2024).

## 6.5 FUTURE WORK

We plan to expand evaluation to larger cohorts and continuous control tasks Chen et al. (2024), integrate advanced models such as self-supervised pretraining Liu et al. (2024) and spiking networks Xu et al. (2021), more dynamic co-adaptation strategies using user-driven labeling, and release MindX as a reproducible benchmark toolkit with standardized APIs and protocols.

Ultimately, MindX aims to foster a reproducible and extensible ecosystem for evaluating real-time BCI systems across cognitive tasks, hardware platforms, and modeling strategies.

## 7 CONCLUSION

We presented MindX, a real-time EEG game control framework using low-cost hardware and lightweight hybrid models. Pilot results show reliable directional decoding, low latency, and encouraging user feedback. The modular design, co-adaptive loop, and open-source release aim to foster reproducible, real-world BCI research. Future work will expand validation, improve game feedback mechanisms, and establish a public benchmark for interactive EEG systems.

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

# A APPENDIX

# A SYSTEM OVERVIEW

## A.1 DUO-STAGE TRAINING PIPELINE

**Duo-Stage Training Pipeline**

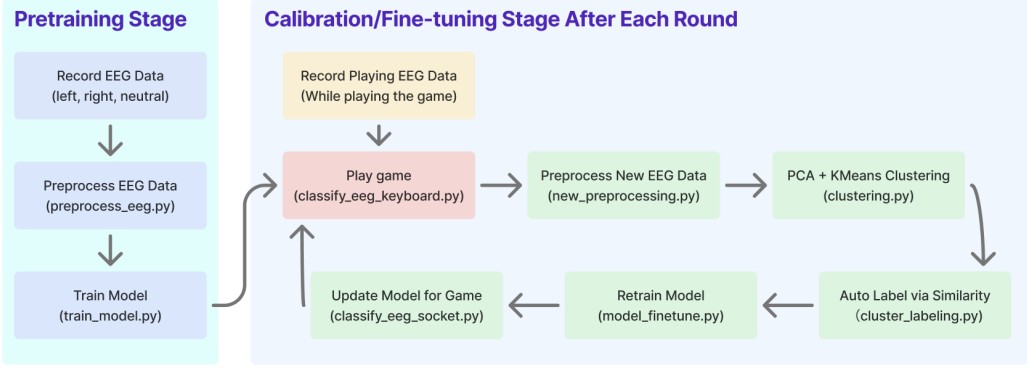

Figure 4: Duo-stage training pipeline for MindX. In the pretraining stage, a supervised model is trained using labeled motor imagery data. In the calibration stage (after each gameplay session), new EEG data is clustered using PCA and KMeans, labeled by similarity to historical data, and used to fine-tune the model.

## B ADDITIONAL METHODOLOGICAL DETAILS

### B.1 SELF-ATTENTION FORMULATION

Given an embedded EEG window $Z \in \mathbb{R}^{T \times d}$, queries, keys, and values are computed as

$$Q = ZW_Q, \quad K = ZW_K, \quad V = ZW_V.$$

The scaled dot-product attention is then:

$$\text{Attention}(Q, K, V) = \text{softmax}\left(\frac{QK^\top}{\sqrt{d_k}}\right) V, \tag{10}$$

where $d_k = d/h$ is the per-head dimension for $h = 2$ attention heads. Attention outputs are pooled across time via mean aggregation.

### B.2 CHANNEL-WISE ATTENTION

$$\alpha_c = \frac{1}{T} \sum_{t=1}^{T} \text{softmax}(w^\top x_{t,c}), \quad \boldsymbol{\alpha} \in \mathbb{R}^C.$$

### B.3 CLASSIFICATION AND CONFIDENCE FORMULATIONS

The model is trained with cross-entropy loss:

$$\mathcal{L}_{\text{CE}} = -\sum_{i=1}^{C} y_i \log(\hat{y}_i). \tag{11}$$

Prediction confidence is quantified using entropy:

$$\mathcal{H} = -\sum_{i=1}^{C} \hat{y}_i \log(\hat{y}_i). \tag{12}$$

Low entropy suggests high confidence, while high entropy reflects uncertainty in the prediction.

### B.4 PSEUDO-LABELING IN ONLINE PERSONALIZATION

Each cluster obtained from PCA and KMeans is assigned a pseudo-label by finding the nearest prior class center:

$$\hat{y}_{\text{new}} = \arg\min_j \text{ dist}(x_{\text{new}}, \mathcal{C}_j). \tag{13}$$

This assignment provides supervision for incremental fine-tuning in the personalization loop.

### B.5 MODEL CONFIGURATION

The model uses:

- Embedding size $d = 32$, two CNN layers, and LDA classifier
- ReLU activations with dropout 0.2
- Optimizer: Adam, learning rate $3 \times 10^{-4}$
- Batch size: 32, Epochs: 20

Experiments were conducted on a Windows laptop (13th Gen Intel(R) Core(TM) i5-13500HX, 16GB RAM) using PyTorch 2.2.1 and real-time data streamed via Lab Streaming Layer (LSL).

## C ADDITIONAL RESULTS

### C.1 QUESTIONNAIRE OUTCOMES

Table 5: Questionnaire results for each of our games using SUS and GEQ

| Game | Snake | Hide and Seek | Rhythm | TUX Racer |
|---|---|---|---|---|
| SUS | 77 | 94.5 | 72.5 | 65.5 |
| GEQ-Competence | 1.9 | 2.63 | 2.3 | 1.0 |
| GEQ-Immersion | 1.31 | 0.57 | 1.93 | 0.65 |
| GEQ-Flow | 0.63 | 2.75 | 2.4 | 1.9 |
| GEQ-Tension | 0.85 | 1.53 | 1.33 | 1.5 |
| GEQ-Challenge | 0.73 | 1.3 | 1.66 | 1.6 |
| GEQ-Negative Affect | 0.8 | 0.75 | 0.95 | 1.7 |
| GEQ-Positive Affect | 2.3 | 2.6 | 2.1 | 2.0 |

