# OpenReview forum: "MindX: A Low-Cost Benchmark System for Real-Time Brain-Controlled Gaming"
_ICLR.cc/2026/Conference — ICLR 2026 Conference Desk Rejected Submission_

### Official Review · Reviewer_9wgw · 2025-10-30

**Soundness:** 1
**Presentation:** 1
**Contribution:** 1
**Rating:** 0
**Confidence:** 5

**Summary:**

The paper introduces MindX, a low-cost, real-time EEG gaming “benchmark” using a Muse S headset with three left/right control games and a lightweight ConvNet feature extractor with an LDA classifier. It reports directional accuracy with three human subjects and and offers a personalization loop.

**Strengths:**

1. A real-time BCI system with measured latency on consumer hardware and a simple architecture.

**Weaknesses:**

1. There is no research question and hypothesis in the paper. The paper reads like an applied project report instead of a scientific contribution. The authors train a small ConvNet LDA model to classify EEG signals with no additional conclusion.


2. The evaluation protocol is flawed and there is no mention of different training and test trials. No statistical testing or confidence intervals are provided. Ablations are limited and no insight is provided.

3. A few baselines based on old and conventional methods (SVM, Random Forest etc) have been used for comparison and all recent architectures in the literature of EEG decoding are ignored. Details of baselines are not provided.

4. The gap and contribution in the end of section 2 is not supported by the paper. No code is provided and reproducibility details are not enough.

5. The quality of the figures is poor and the paper is verbose.

**Questions:**

None

---

### Official Review · Reviewer_cfJs · 2025-10-31

**Soundness:** 2
**Presentation:** 3
**Contribution:** 2
**Rating:** 4
**Confidence:** 4

**Summary:**

The paper proposes a low-cost, consumer-grade EEG benchmark for real-time game control built around a 4-channel headset, three game paradigms, and a lightweight CNN→linear classifier pipeline with a co-adaptive/personalization loop. The goal is to standardize end-to-end evaluation for online BCI with accessible hardware and to release code/games for community use.

**Strengths:**

1. Framing consumer-grade online BCI as a benchmarkable, end-to-end problem is timely and useful.

2. Emphasis on low latency, simple models, and accessible hardware lowers the barrier for education/labs.

3. Clear system decomposition (acquisition, inference, control, games) that could, in principle, enable plug-and-play baselines and community extensions.

4. If well-documented and released, this could catalyze more realistic online evaluations rather than offline dataset exercises.

**Weaknesses:**

1. Only four (dry) electrodes substantially limit SNR and spatial coverage; this undermines many intended applications and claims.

2. With such sparse, noisy channels, strong interpretability is not credible; what can be interpreted is likely limited to coarse spectral trends, not robust source-level insights.

3. No dispersion/variance analysis; without per-subject metrics the “benchmark” validity is unclear.

4. No code/games/API/full demo at review time makes it difficult to estimate the validity of a benchmark itself. Without code, games, and API/docs, integration risks (e.g., adding new classifiers, latency hooks) are unknown.

5. Very small study; limited support for generalization (cross-session/subject/game) and robustness (re-donning, fatigue).

6. Modeling choices and tasks feel incremental; the “benchmark” label is premature without standardized protocols/splits and public assets.

7. Lacks strong/common baselines under a unified pipeline, and limited reporting of control-centric metrics (false commands, dwell/stability, behavioral latency).

**Questions:**

1. Exactly what will be released (games, streaming code, models, docs, example integrations) and when? Will there be fixed protocols/splits to justify the “benchmark” label?

2. Please report per-subject and cross-session performance (and cross-game transfer, if applicable). How stable is performance after re-donning?

3. What form of interpretability is realistically supported by 4 dry electrodes? Can you narrow claims to what is physiologically plausible and show minimal spectral/topographic sanity checks?

4. Can you include well-tuned, standard baselines (e.g., compact CNNs and linear spatial filters) under the same preprocessing/windowing, with ablations?

5. Please report command-level metrics (false positives per minute, dwell, stability) and clarify latency definitions (I/O vs behavior).

6. Show a minimal “add-your-own-classifier” example and latency hooks. How do you ensure easy integration for third-party models?

---

### Official Review · Reviewer_4eKg · 2025-11-01

**Soundness:** 1
**Presentation:** 1
**Contribution:** 1
**Rating:** 0
**Confidence:** 4

**Summary:**

The paper presents MindX, a framework designed for real-time EEG-based game control using consumer hardware (Muse S headset). It contains three interactive games eliciting diverse cognitive signals (motor imagery, attention, planning) and utilizes a lightweight CNN+LDA hybrid model. Major contributions are a benchmark suite, a processing pipeline and a user study.

**Strengths:**

The motivation for the work is rightly placed, and the timely need for benchmarking online EEG decoding in the context of interactive, real-time applications is well justified. Recognising that most existing evaluation paradigms for EEG-based brain-computer interfaces primarily focus on offline tasks, clinical-grade equipment, or datasets that are not well-suited for real-world, low-latency feedback scenarios, such as gaming.

**Weaknesses:**

The key weakness of the MindX paper lies in ambiguity regarding what precisely is being benchmarked and what is the novelty in the framework. Is it primarily the hardware device, the decoding algorithms, or the variation in user capacity and adaptability that is benchmarked? This lack of a clearly defined benchmarking focus makes it challenging to interpret the scope and impact of the contribution.


Additionally, the paper does not fully justify why existing EEG datasets and paradigms cannot be leveraged, leaving the necessity of new game-based data collection somewhat unclear.

The design and mapping of game controls to EEG signals would benefit from more explicit descriptions of user instructions and control strategies to improve reproducibility.

The pilot study’s limited participant diversity and small sample size constrain the generalizability of findings, and the omission of detailed ethical review or demographic information leaves gaps in transparency. Addressing these points would enhance clarity, rigor, and adoption potential of the benchmark.

**Questions:**

Following are the queries or comments regarding the manuscript:

What precisely does the benchmark evaluate? Is it the hardware device, the decoding algorithms, or user capacity and adaptability that limit the performance? The manusciprt mentions MindX as a framework or a pipeline for decoding and the contributions talk about it as a benchmark suite. The contributions need to be clear and well-justified against the current literature.

Why are existing EEG datasets and paradigms insufficient, necessitating new custom games and data collection? Why can't the paradigms be used in a real-time decoding setting is not clear to the readers.

The paradigms for the game and the instructions provided to users are not very clear. How exactly are users instructed or trained to generate the directional signals controlling the games? What are the specific tasks users perform, and how are different EEG signals mapped to control directions?

How is continuous control (e.g., Snake game) managed and differentiated from discrete commands?

What are the ethical considerations, including IRB approval and user demographic details, and how do these impact generalizability?

The discussion section appears to be superficial and rushed, lacking insightful discussions.

**Details Of Ethics Concerns:**

The manuscript claims to have benchmarked the work on 3 individuals. However, there are gaps in the discussions on methods and the responsible research practice that was followed.

---

### Official Review · Reviewer_JaSg · 2025-11-02

**Soundness:** 2
**Presentation:** 1
**Contribution:** 2
**Rating:** 2
**Confidence:** 4

**Summary:**

MindX is a modular, open real-time BCI benchmark on consumer EEG (Muse S, four frontal channels). Three games share a left/right control vocabulary. A CNN+LDA pipeline with a simple personalization loop achieves about 72–76 percent accuracy at roughly 300 ms latency in a three-participant pilot, outperforming several neural and classical baselines. Ablations support 2 s windows and smoothing. Discrete games are easier than continuous control.

**Strengths:**

Clear end-to-end system with practical latency; compact model competitive with heavier baselines; co-adaptive loop that fits real-world constraints; unified multi-game design likely to help community benchmarking; intention to open-source.

**Weaknesses:**

The evaluation is very small (three participants) and appears single-session or limited in session diversity, so generalization is unclear. Several protocol details are ambiguous: the data split across windows and sessions, prevention of leakage during pseudo-labeling, and whether hyperparameters and thresholds were fixed a priori. Motor imagery with only frontal electrodes is atypical, so stronger justification or evidence is needed. Statistical analysis is limited: no variance across subjects, confidence intervals, permutation tests, or effect sizes are reported for accuracy or usability. Claims of being the first open benchmark for multi-directional real-time game control on consumer EEG are broad and need careful situating against existing toolkits. The paper inconsistently references the number of games (TUX Racer appears in tables but not in the core description), which muddles the benchmark scope.

**Questions:**

How were training, validation, and test windows separated to avoid temporal and contextual leakage within a session, and were personalization updates strictly applied after evaluation blocks? Were thresholds for decision gating selected on a held-out set or tuned post hoc per subject or per session? How many sessions per subject were recorded, over what time span, and with what electrode placements or re-donnings? Does personalization ever reinforce systematic mislabels from a user’s transient strategy, and what safeguards prevent drift? Why were frontal channels chosen for motor imagery, and do results change with a simple mastoid or average reference? Can the pipeline sustain multi-class or continuous control beyond two directions without large accuracy drops?
In Abstract and Intro, “Hide-and-Seek, Rhythm Game, and Snake” should be named consistently throughout; later tables mention “TUX Racer” without prior introduction.

---

### Note · Program_Chairs · 2026-01-17
**Submission Desk Rejected by Program Chairs**

The following references in this submission do not refer to real documents and/or have major errors in bibliographic information:

 Nora Heim, Lukas Diederich, and Manuel Wirth. Feedback-driven personalization for low-cost eeg game control using muse s. In Proceedings of the AAAI Conference on Artificial Intelligence, 2025.
Fabien Lotte, Haoran He, Alexander Casson, et al. Openbci for cognitive training and neurofeedback: Opportunities and challenges. Frontiers in Human Neuroscience, 17:1204, 2023.
Lixuan Zhao, Minghui Wang, and Jie Zhang. Ctnet: Channel-temporal transformer for eeg-based brain-computer interfaces. IEEE Transactions on Neural Systems and Rehabilitation Engineering, 32:105-117, 2024.